# Diversity-triggered deterministic bacterial assembly constrains community functions

Weibing Xun[1,2], Wei Li[1], Wu Xiong[1], Yi Ren[1], Yunpeng Liu[2], Youzhi Miao[1], Zhihui Xu[1], Nan Zhang[1], Qirong Shen[1] & Ruifu Zhang[1,2]

A growing body of evidence suggests that microbial α-diversity (local species richness) may have positive effects on ecosystem function. However, less attention has been paid to β-diversity (the variation among local microbial assemblages). Here we studied the impact of microbial α-diversity on stochastic/deterministic microbial community assembly processes, which are related to β-diversity, and the consequences for community function. Bacterial communities differing in α-diversity were generated and their structures and potential community functional traits were inferred from DNA sequencing. Phylogenetic null modeling analysis suggests that stochastic assembly processes are dominant in high-diversity communities. However, in low-diversity communities, deterministic assembly processes are dominant, associating with the reduction of specialized functions that are correlated with specific bacterial taxa. Overall, we suggest that the low-diversity-induced deterministic community assembly processes may constrain community functions, highlighting the potential roles of specialized functions in community assembly and in generating and sustaining the function of soil ecosystems.

[1] Jiangsu Provincial Key Lab of Solid Organic Waste Utilization, Jiangsu Collaborative Innovation Center of Solid Organic Wastes, Educational Ministry Engineering Center of Resource-Saving Fertilizers, Nanjing Agricultural University, Nanjing 210095, China. [2] Key Laboratory of Microbial Resources Collection and Preservation, Ministry of Agriculture, Institute of Agricultural Resources and Regional Planning, Chinese Academy of Agricultural Sciences, Beijing 100081, China. Correspondence and requests for materials should be addressed to Q.S. (email: shenqirong@njau.edu.cn) or to R.Z. (email: rfzhang@njau.edu.cn)

Microbial communities are the most diverse and dominant groups of organisms in terrestrial ecosystems. The activity of microorganisms greatly influence a variety of ecosystem functions and contribute to soil productivity and nutrient cycling, as well as many other ecosystem properties and services[1]. One major general hypothesis states that ecosystems containing numerous species exhibit high levels of ecosystem functions and are functionally stable[2]. This prediction assumes that species overlap can sufficiently provide some redundancy and buffer ecosystem functions against biodiversity loss[3]. An extensive body of theoretical, experimental, and observational studies across different types of ecosystems confirm that high level of local diversity is important for generating and sustaining ecosystem functions and services[4–6]. Thereby, a dramatic loss of biodiversity has significant adverse impacts on ecosystem functions[7]. However, other attempts to study the relationships between biodiversity and ecosystem function have yielded mixed results[8–12]. Therefore, although high levels of bacterial diversity have led to the assumption that some bacteria are functionally redundant, we know little concerning how the interactions within diversity-related bacterial communities mediate ecosystem function and community assembly.

Various driving factors of microbial diversity have been intensively investigated across different spatial and temporal scales[13–15]. Environmental surveys demonstrate that soil bacterial diversity can be strongly affected by pH, soil type, latitude, vegetation, moisture, temperature, and nutrient availability. Among these, soil pH is the best predictor of both soil bacterial diversity and richness[16], whereas soil type strongly influences soil bacterial composition[17]. Although the general patterns underlying variations in biodiversity have been observed, the factors controlling these patterns remain unclear. In general, the generation of soil microbial diversity and function is referred to as the community assembly processes. Microbial community assembly, which reflects the aggregation of spatiotemporal processes[18] that determine community composition, falls into two predominant categories that can be summarized as deterministic and stochastic processes[19]. Regardless of which of these two processes is dominant, community assembly determines the presence and abundance of species. According to Wardle and Putten[20], the species abundance is also an important determinant of ecosystem functions and the identity of those species. Therefore, the assembly process can necessarily influence soil microbial diversity and composition, with downstream impacts on the function of the system[21]. Mori et al.[22] recently point out that β-diversity, species turnover between two sites or communities, is especially important in the context of ecosystem multifunctionality. However, much less attention has been paid to β-diversity than to α-diversity. β-diversity is useful for inferring the stochastic and deterministic processes of community assembly for soil microbial communities[23]. Therefore, deeper knowledge of soil microbial β-diversity and the linkage between diversity and function will lead to a better understanding of the contributions of biodiversity to functional processes.

In this study, we establish a soil microcosm incubation experiment by inoculating progressively diluted soil suspensions into sterilized and pH-amended soils, to clarify the uncertain relationship between α-diversity and assembly processes (stochastic/deterministic processes related to β-diversity), and to assess the role of assembly processes in generating community function. We present the results of taxonomic features and potential community functional traits of the re-assembled bacterial communities from DNA sequencing and find that the reduced specialized functions dominate the deterministic community assembly processes, laying a foundation for deciphering the linkages between biodiversity and community functions.

## Results

**Taxonomic features of the re-assembled bacterial community.** Dilution had significant impacts on the composition and α-diversity of the re-assembled bacterial communities (Fig. 1a, b and Supplementary Fig. 1). The enriched bacterial groups related to high dilution levels were significantly ($P$-value < 0.001, one-way analysis of variance (ANOVA)) over-represented in the phyla/class of *Betaproteobacteria*, *Gammaproteobacteria*, and *Bacteroidetes*, whereas the depleted bacterial groups were mostly affiliated with *Acidobacteria*, *Actinobacteria*, *Nitrospira*, and *Deltaproteobacteria*. Bacterial α-diversity, as expressed by the Shannon diversity index, was the highest in the initial soils and decreased along the dilution gradient (Fig. 1b). Although the suspensions were the same when inoculated into the two types of pH-amended soils, bacterial α-diversity peaked at neutral pH. However, these diversity indices were less variable across pH levels than across dilution levels. Non-metric multidimensional scaling (NMDS) analysis based on Bray–Curtis dissimilarity (Fig. 1a) showed that the samples derived from black soil grouped separately from the samples derived from red soil on the horizontal axis. Moreover, the samples across the soil pH gradient could be distinguished on the vertical axis, with samples being more dispersed at higher dilution levels.

Taxonomic abundance-based Weighted UniFrac community dissimilarity between different pH levels showed significantly larger variances within the more diluted treatments for both black and red soils (Fig. 2a), indicating larger variances within the communities that exhibited lower α-diversity. Large variances between the two types of soils were observed at every dilution level (Fig. 2b), although these variances became smaller within more diluted samples at the same pH level (Supplementary Fig. 2).

Although soil type and pH had less significant impacts on soil bacterial α-diversity than dilution, they nevertheless affected the structure of re-assembled bacterial community. The relative contributions of soil type, soil pH, and the concentrations of Ca, Fe, and $SO_4^{2-}$ on the community variances were evaluated using variation partitioning analysis (VPA) for each dilution level (Fig. 2c). The contribution of soil type to the community variances decreased from 46.2% ($P$-value < 0.001 based on partial mantel test) in the initial soils (Fig. 2d) and 47.9% ($P$-value < 0.001 based on partial mantel test) in the $10^{-1}$ diluted samples (Fig. 2e) over 40.2% ($P$-value < 0.001 based on partial mantel test) in the $10^{-4}$ diluted samples (Fig. 2f) to 29.7% ($P$-value < 0.001 based on partial mantel test) in the $10^{-7}$ diluted samples (Fig. 2g) and 19.3% ($P$-value < 0.001 based on partial mantel test) in the most diluted samples (Fig. 2h). In contrast, the effect of soil pH increased along the dilution gradient (9.6%, 10.7%, 13.6%, 28.1%, and 43.3% of the variance in the initial soil and increasingly diluted samples, respectively; $P$-value < 0.001 based on partial mantel test). Moreover, the concentrations of Ca and Fe had little impacts on the re-assembled communities, although different amounts of CaO and FeSO₄ were used to manipulate soil pH. Consistently, permutational multivariate ANOVA (PERMANOVA) showed that soil type ($P$-value = 0.001, $F_{1,502} = 16.73$ using PERMANOVA) and pH ($P$-value = 0.001, $F_{4,475} = 7.02$ using PERMANOVA) altered the composition of bacterial communities significantly, explaining 35.2% and 29.5% of the variances, respectively. Meanwhile, the concentrations of Ca, Fe, and $SO_4^{2-}$ explained only 3.0% ($P$-value = 0.037, $F_{9,470} = 1.97$ using PERMANOVA), 2.4% ($P$-value = 0.063, $F_{9,470} = 1.49$ using PERMANOVA), and 1.1% ($P$-value = 0.046, $F_{9,470} = 1.90$ using PERMANOVA) of the variances in bacterial community composition, respectively.

**Assembly processes of the bacterial communities.** To discriminate between the deterministic and stochastic processes in

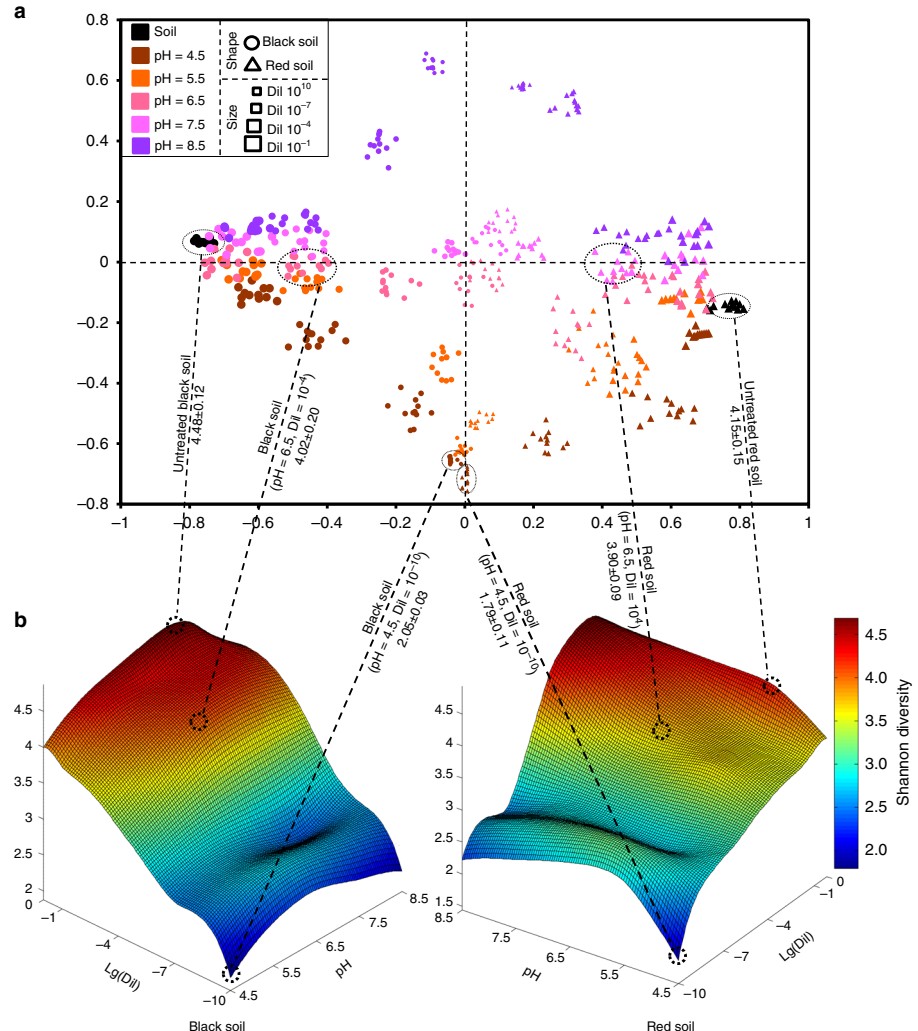

**Fig. 1** Dimension reduction analysis and Shannon diversity of re-assembled bacterial communities. **a** Non-metric multidimensional scaling (NMDS) analysis of all incubated soil samples based on the Bray–Curtis taxonomic similarity. **b** The variances of Shannon diversity indices across soil pH and dilution gradients for black (left) and red (right) soils. Selected samples and corresponding diversity indices are connected by dotted lines. Lg(Dil) indicates the Lg-transformed dilution level. Lg(Dil) = 0 represents the untreated soil

community assembly along the dilution and pH gradients, we calculated the β-nearest taxon index (βNTI) of every treatment. To infer alterations in the deterministic/stochastic assembly processes along dilution and pH gradients, we examined the relationships between βNTI and dilution and pH gradients. The βNTI values within each treatment showed various patterns related to pH and dilution level (Fig. 3). In both black and red soils, community assembly shifted from stochastic processes (|βNTI| < 2) to deterministic processes (|βNTI| > 2) with increasing dilution levels and the βNTI values were significantly correlated with dilution gradients for all pH levels (Fig. 3a, b). However, the trends in βNTI distribution over dilution gradients were not identical among different soil pH levels. Stochastic assembly processes were dominant in the least diluted samples and shifted towards dominance of variable selection (βNTI > 2) at the pH value of 6.5, while there was a shift towards dominance of homogeneous selection (βNTI < −2) under more acidic (pH 4.5) and alkaline (pH 8.5) conditions in more diluted samples.

The same βNTI datasets were reorganized to examine βNTI distributions across the pH gradient from 4.5 to 8.5 for each dilution level. For both the black and red soils, the βNTI values exhibited unimodal patterns along the pH gradient (Fig. 3c, d). Peak values were observed for neutral pH soils, whereas valley

values were observed under acidic and alkaline conditions. In addition, the magnitude of these patterns increased with higher dilution levels. These results suggested that a bacterial community with lower species richness was more likely to follow deterministic assembly processes. Lastly, all βNTI values were combined, resulting in a significantly negative relationship between |βNTI| and soil bacterial Shannon diversity indices (Spearman's correlation coefficient $R^2 = 0.38$, P-value < 0.001, two-sided tests; Fig. 3e).

**Potential functions of the re-assembled bacterial community.** To assess how soil pH and dilution affect the potential functions of the re-assembled bacterial communities, shotgun metagenomic sequencing and analyses were conducted on the DNA extractions from the red soil samples. We compared substance metabolism-related genes, most of which are related to carbon, nitrogen, and sulfur turnover, at various pH and dilution levels (Fig. 4a). These functional categories were chosen, as many ecological services, e.g., food production and pollutant degradation and purification, rely largely on microbial functions related to the metabolism of varying substances. The relative abundances of every functional category differed significantly among dilution levels but less significantly, or in some cases not at all, across different pH levels.

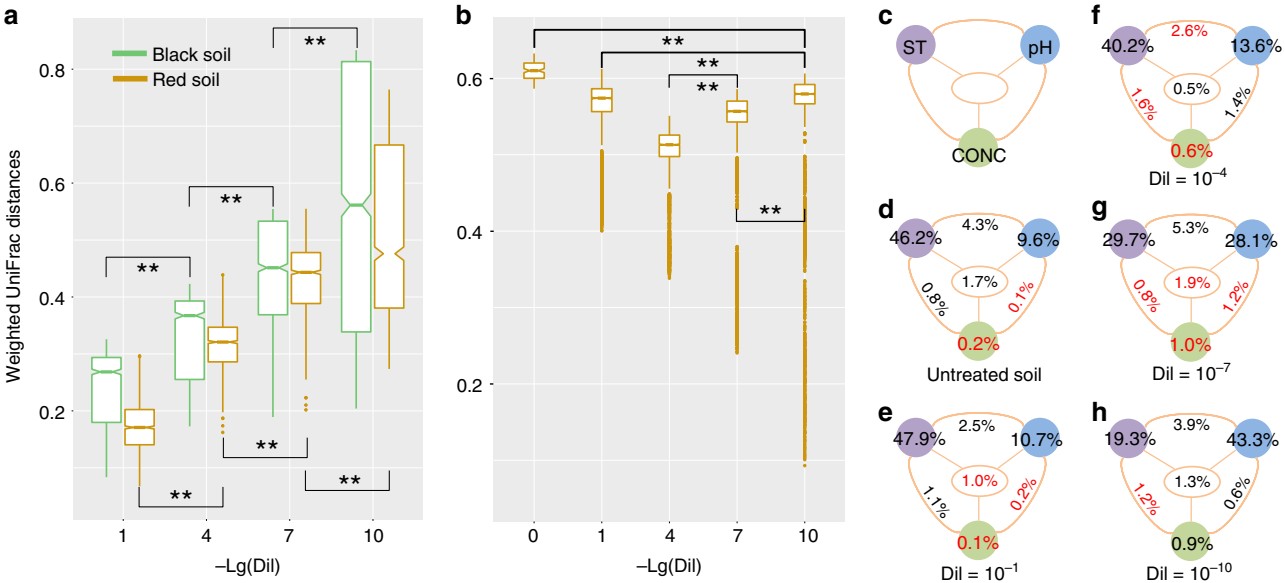

**Fig. 2** Bacterial community variations between and within soil types. **a** Weighted Unifrac distances of pairwise bacterial communities from different pH levels within black and red soils, respectively, at each dilution level. **b** Weighted Unifrac distances of pairwise bacterial communities at different dilution levels between black and red soil samples. Lg(Dil) indicates the Lg-transformed dilution level. Lg(Dil) = 0 represents the untreated soil. Asterisks indicate significance: **$P$-value < 0.01 based on Tukey's HSD test. Boxplot: median, 25%/75% percentiles, and the highest, lowest, and extremely values are shown. **c** Variation partitioning analysis (VPA) among soil type (ST), soil pH (pH), and the concentrations of Ca and Fe (CONC) for **d** untreated soil, and **e** $10^{-1}$, **f** $10^{-4}$, **g** $10^{-7}$, and **h** $10^{-10}$ diluted samples. Numbers indicate the percentage of variations in the bacterial community. Black numbers indicate significant variances ($P$-value < 0.01 based on Mantel test). Red numbers indicate variances that are not significant

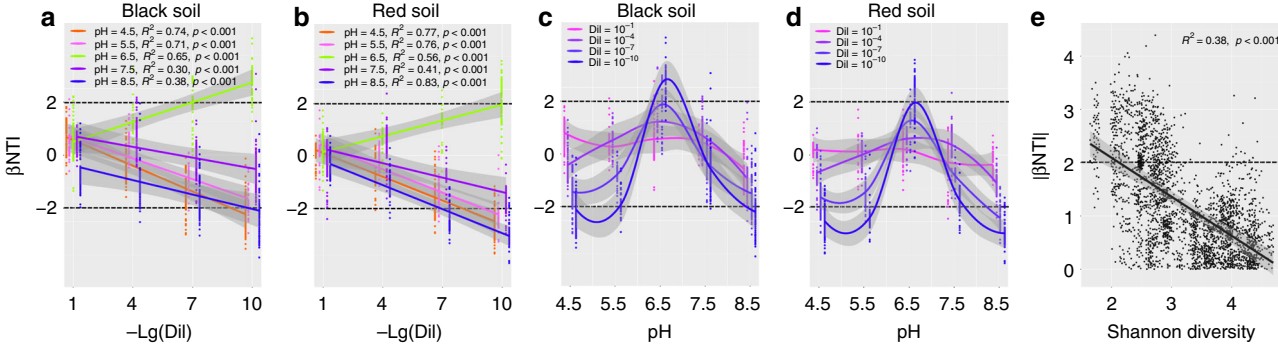

**Fig. 3** βNTI values are associated with dilution and pH levels, and bacterial Shannon diversity. **a** The relationships between −Lg(Dil) and βNTI in black soil. **b** The relationships between −Lg(Dil) and βNTI in red soil. **c** The effect of soil pH on βNTI in black soil. **d** The effect of soil pH on βNTI in red soil. **e** The relationship between |βNTI| and bacterial Shannon diversity indices. Dil indicates the dilution level. Lg(Dil) indicates the Lg-transformed dilution level. Lg (Dil) = 0 represents the untreated soil

A function carried out by narrow groups of microorganisms, meaning the related genes are distributed in specific microorganisms, will become gradually less abundant in samples seeded with more diluted inocula. For instance, the overall functions (specialized functions defined as functional group 1, FunGp1 hereafter) related to "Sulfur metabolism," "Nitrogen metabolism," "Methane metabolism," "Terpenoids & Polyketides metabolism," and "Xenobiotics biodegradation and metabolism" (including "Atrazine degradation" and "Polycyclic aromatic hydrocarbon degradation," etc.) were gradually eliminated as the dilution increased (Fig. 4a). Correlation analysis of all the unique genes within these categories demonstrated that large proportions (31.45–64.87%) of these genes were significantly decreased in relative abundances along the dilution level, whereas only small proportions (1.67–9.22%) of these genes were significantly increased (Supplementary Table 1). Among these genes, indicator analysis (IndVal) showed that a specific gene of FunGp1, the periplasmic nitrate reductase gene *napA* of nitrogen metabolism

(IndVal = 0.938, $P$-value = 0.001 using the multipatt function of IndVal), was enriched as an indicator gene in more diluted samples, whereas other genes, such as phosphoadenosine phosphosulfate reductase gene *cysH* of sulfur metabolism (IndVal = 0.946, $P$-value = 0.002 using the multipatt function of IndVal) were depleted by dilution (Supplementary Table 2).

In contrast, other functions (broad functions defined as functional group 2, FunGp2 hereafter) related to "Glycolysis/ Gluconeogenesis," "TCA cycle," and the metabolism of some readily usable carbohydrates such as fructose, galactose, starch, and sucrose, were more presented in more diluted samples (Fig. 4a). For these functional categories of FunGp2, large proportions (34.89–61.95%) of these genes were significantly increased with dilution level, whereas only small proportions (1.98–8.12%) of these genes were significantly decreased (Supplementary Table 1). Indicator analysis showed that the metabolic genes of some essential amino acids (phenylalanine and tryptophan) were depleted, whereas the metabolic genes of some non-essential

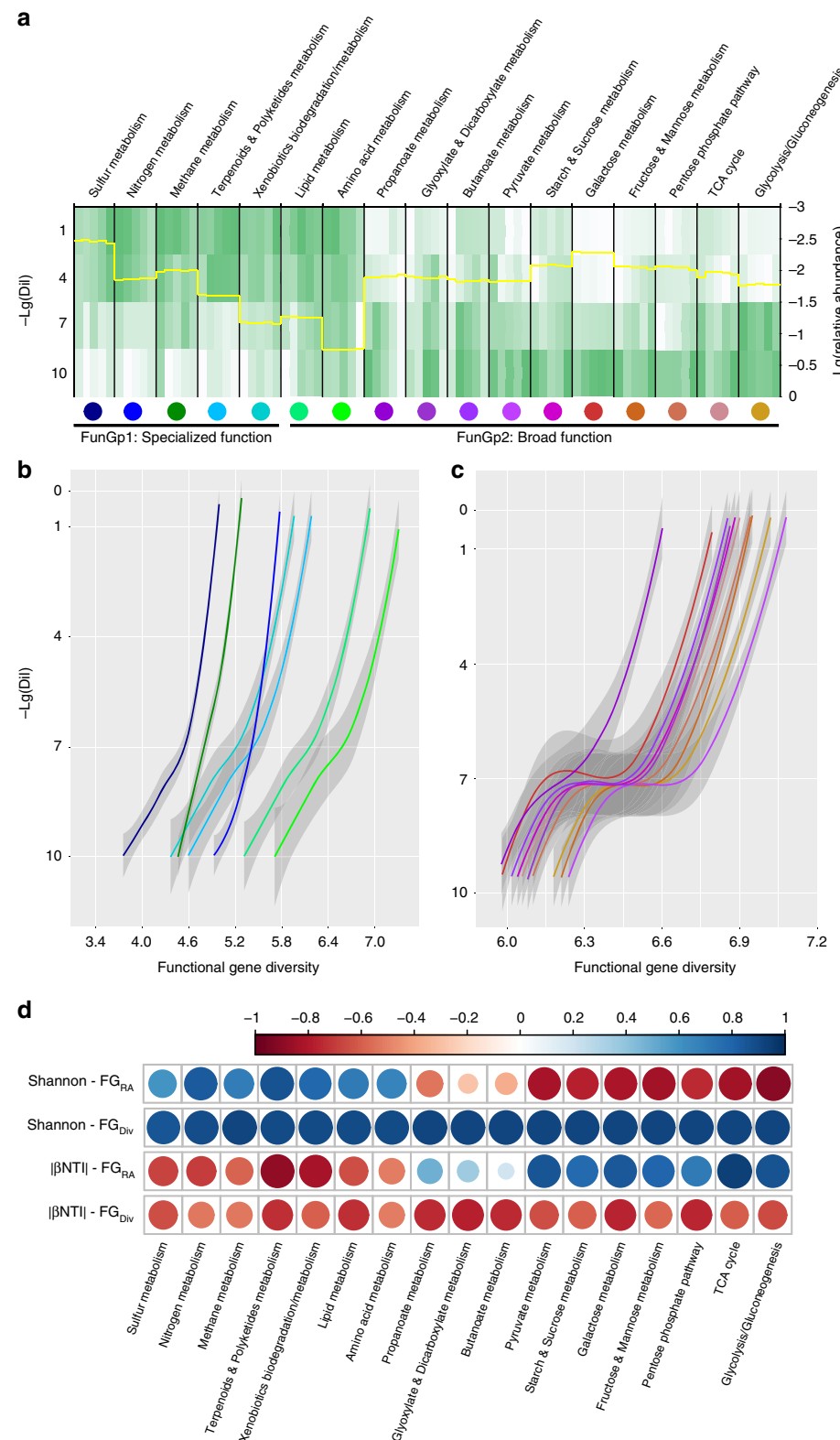

**Fig. 4** Functional genes differ in abundance and diversity among treatments. **a** Heatmap of the relative abundances of different functional categories based on the normalized metagenomic data in $10^{-1}$, $10^{-4}$, $10^{-7}$, and $10^{-10}$ diluted samples. Color scaled from highest (green) to lowest (white) relative abundances within functional categories. Five columns of each functional category in heatmap indicate the soil pH range from 8.5 (left) to 4.5 (right). Yellow solid line indicates the Lg-transformed relative abundances of various functional categories. **b** The effect of dilution on specialized functional gene diversity. **c** The effect of dilution on broad functional gene diversity. The color lines in **b** and **c** represent different functional categories indicated by the color points in **a**. Lg(Dil) indicates the Lg-transformed dilution level. Lg(Dil) = 0 represents the untreated soil. **d** Correlations between Shannon and $FG_{RA}$, Shannon and $FG_{Div}$, $|\beta NTI|$ and $FG_{RA}$, and $|\beta NTI|$ and $FG_{Div}$. Shannon: bacterial Shannon diversity. $FG_{RA}$: the relative abundance of a detected functional category. $FG_{Div}$: the diversity of a detected functional category. The color points in **d** indicate the correlation coefficient (bluer indicates stronger positive correlation and redder indicates stronger negative correlation) and significance (larger size indicates lower $P$-value)

amino acids (histidine) were enriched under dilution (Supplementary Table 2). However, not all functional categories in FunGp2 follow this trend; the "Amino acid metabolism" and "Lipid metabolism" functions showed a general depletion trend by dilution with 36.76% and 30.09% of significant decreased genes, respectively (Fig. 4a and Supplementary Table 1).

Despite of these opposing patterns on the relative abundances of FunGp1 and FunGp2, the diversity of all functional categories followed a decreasing pattern with increasing dilution level (Fig. 4b, c). Moreover, the decreasing amplitudes of functional diversity were greater for FunGp1 (Fig. 4b) when compared with FunGp2 (Fig. 4c). To further investigate the impact of bacterial diversity and deterministic/stochastic assembly processes on the relative abundance ($FG_{RA}$) and diversity ($FG_{Div}$) of metabolic functions, we calculated the correlations between $|\beta NTI|$ values and their corresponding Shannon diversity indices, $FG_{RA}$ values, and $FG_{Div}$ values (Fig. 4d). The diversity and relative abundance of functional traits belonging to FunGp1, as well as the diversity of functional traits belonging to FunGp2, were positively correlated with Shannon diversity and negatively correlated with $|\beta NTI|$ values. In contrast, the relative abundances of functional traits belonging to FunGp2 were negatively correlated with Shannon diversity, but positively correlated with $|\beta NTI|$ values. Therefore, FunGp1 and FunGp2 responded differently to bacterial diversity loss and may contribute differently to bacterial community assembly.

## Discussion

The method used in this study was based on the inoculation of serially diluted soil suspensions into γ-irradiated sterilized and pH-amended soil samples, to manipulate the bacterial species richness of the re-assembled microbiome after microcosm incubation[24,25]. Our objective was to assess the importance of bacterial diversity-related functional gene pool in establishing a community assemblage. In the samples with the same dilution level, we only noticed a moderate shift in the bacterial species richness across soil pH levels. Shannon diversity indices peaked at neutral pH and were similar or, in some cases, significantly higher in black soil than in red soil. These differences were likely due to the direct impact of pH on community diversity[16] and might be related to the higher nutrient availability in black soil[26]. These relatively moderate shifts in bacterial diversity of re-assembled microbiome correlated to soil type and pH were contrasted to the much stronger dilution effects. Therefore, dilution reduced the bacterial diversity in the soil suspensions used as inocula[24] and subsequently affected the bacterial diversity in the re-assembled soil microbial communities, which confirmed the results of previous studies[25].

We observed consistently significant positive correlations between bacterial α-diversity and functional gene diversity. Functional gene categories of FunGp1 and FunGp2 responded to dilution differently. The differences were particularly striking for the genes encoding functions related to "Sulfur metabolism," "Nitrogen metabolism," "Methane metabolism," "Terpenoids & Polyketides metabolism," and "Xenobiotics biodegradation and metabolism" of FunGp1, which were depleted upon dilution, although a small proportions of these genes were not. The influence of taxonomic diversity decline on the functional genes with increasing dilution may be explained by the lack of taxonomic redundancy for these specialized functions[27]. Due to their high dependency on narrowly distributed physiological pathways[28], these specialized functions are generally assumed to be highly dependent on environmental conditions[29]. Due to their scarcities in most soil microbiome, these specific functions would be particularly affected by dilution, as they are more likely to be removed over more abundant taxa. Numerous studies have suggested that some bacterial members of *Acidobacteria*, *Actinobacteria*, and *Nitrospira* are keystone taxa, which may harbor specialized functions such as nitrogen fixation or ammonia oxidation[30–32]. For instance, the gene clusters inferring to "Terpenoids and Polyketides metabolism" are usually harbored by a minority of soil microbial community, such as some species of *Actinobacteria*[33]. Several investigations have demonstrated that the enzymes involved in "Xenobiotics biodegradation/metabolism" are inducible and only exist in a small group of microorganisms[34,35]. Consistently, the taxonomic composition results in our study demonstrated that the bacterial groups of *Acidobacteria*, *Actinobacteria*, and *Nitrospira* were depleted in more diluted samples. Thus, the dilution-induced loss of rare species may significantly constrain the physiological pathways of specialized functions and result in a weakened metabolic network.

The functions of FunGp2, related to "Glycolysis/Gluconeogenesis," "TCA cycle," and "Pentose phosphate pathway" are critical for microbial growth and propagation. These functions, the so-called broad functions[27], are widely distributed among living organisms and were enriched by dilution. Among these broad functions, the "Amino acid metabolism" and "Lipid metabolism" functions are quite widely distributed. For instance, the biosynthesis of iso- and anteiso-fatty acids occurs in many bacteria, as the major acyl constituents of membrane lipids[36] and the biosynthesis pathways of non-essential amino acids[37] are distributed broadly among diverse bacteria. However, these two functions were generally depleted by dilution. We inferred that the biosynthesis of omega-cyclohexyl and omega-cycloheptyl fatty acids[36], and the metabolisms of some essential amino acids[38] are only present in small populations of bacterial species. Consequently, the "Amino acid metabolism" and "Lipid metabolism" are functions in between the specialized and broad functions. Moreover, we observed that the diversity of these two functional groups decreased with increasing dilution levels. Besides, the decreasing amplitudes in FunGp2 were smaller than those in FunGp1. These observations suggested that the broad functions were more redundant than the specialized functions and the overlap of the functional capabilities of species in a community may follow different subtractive shapes[39].

Functional gene diversity is an important indicator for functional redundancy, whereas gene abundance may represent functional capacity. Soil microbial diversity loss results in a significant decrease in specialized functional capacity, such as potential denitrification activity[40] and pesticide mineralization capacity[34]. In contrast, with the inoculation of gradient-diluted soil suspensions in an established microcosm incubation experiment, Griffiths et al.[12] found there were no differences in soil respiration, which is a broad function, although the microbial diversity decreased with intensified dilution. Therefore, we suggest that the decreasing of specialized functions synchronously resulted in increased proportions of broad functions and, hence, broad functions were more abundant in more diluted samples.

The relationships between decreases in functional gene diversity and the degradation of ecosystem function and service have been widely investigated based on α-diversity, but particularly lacking on β-diversity[22]. Considering there is no ubiquitous species assemblage that can support all functions simultaneously, community functions require different sets of species assemblages under various environmental conditions[41]. Moreover, the stochastic/deterministic processes to form species assemblages maybe well-defined by β-diversity[23]. Consequently, β-diversity-related analyses should be considered in studies investigating relationships between biodiversity and ecosystem function.

In this study, for each type of soil, the bacterial communities from less diluted samples were more similar than those from

more diluted samples. Dilution reduced species richness, leading to decreased resilience and increased sensitivity to soil pH changes. As community stability is enhanced in species-rich communities[42–44], a community with lower species richness will be more susceptible to environmental changes, e.g., soil pH changes. Not surprisingly, the VPA results demonstrated that the effect of soil type on the re-assembled bacterial communities decreased, whereas the effect of pH increased with increasing dilution levels. However, we observed few significant differences in the abundances of every functional category among soil pH levels, suggesting that the community functional gene composition may not be constrained by taxonomy[45].

Our results showing the shift patterns of βNTI values through different pH and dilution levels clearly indicate that the community assembly tends to be dominated by deterministic processes when the taxonomic diversity level is low. This tendency was further confirmed by calculating the relationships between |βNTI| values and bacterial Shannon diversity indices. Previous studies[46,47] have suggested that the community is more susceptible to drift (stochastic process) or founder effects with lower biomass and smaller population. This is true at the period of low population or community size[48]. However, in an established community with saturated population or community size, the local dominance of stochastic or deterministic processes can be strongly affected by dispersal, which depends on the local environment[49]. In addition, we found the deterministic processes were correlated with decreased functional diversity and relative abundances of specialized functions. Hence, we suggest that dispersal in low-diversity communities may enhance environmental selection and contribute to deterministic (selection) processes. Therefore, there is a transition in bacterial community assembly from stochastic to deterministic processes with the decreasing of soil bacterial species richness and functional diversity. Moreover, according to the βNTI values, deterministic processes tend to be a variable selection (βNTI > 2) at neutral pH and a homogeneous selection (βNTI < −2) under acidic and alkaline conditions.

Taken together, this study has collectively visualized the structural and functional assembly patterns of bacterial communities on broad species richness across soil pH values in two types of soils. We therefore integrated these results into a conceptual model on how diversity-related stochastic/deterministic processes contribute to the structural and functional assembly of soil bacterial communities under various environmental conditions (Fig. 5). We assumed a re-assembled community in which the relative influences of stochastic/deterministic ecological processes become relatively stable in this framework. However, these stable processes may not exist in real ecosystems due to changing environments. We also did not take into account of the initial stage of community establishment, which is primarily dominated by stochastic processes[50]. However, in the later stages of assembly, microorganisms need to cope with two main stressors: environmental selection and species competition[51].

Part I: The assembly of a high-diversity community. In this stage, community assembly is probably driven by comprehensive biotic and abiotic impacts[52]. Due to the high level of bacterial diversity, assembly is expected to involve complex biological interactions[53,54] and nutrient availability is responsible for the intensity of these interactions[55]. Soil contains large amounts of complex and recalcitrant substrates that cannot be efficiently utilized by most microbes[56] and the competition will intensify when available nutrients in the environment are insufficient to support a large number of microbes[57]. Consequently, specialized functions, including "Sulfur metabolism," "Nitrogen metabolism," etc., can maintain nutrient turnover in the soil and provide available nutrients for other microorganisms, thus potentially weakening the biotic competition. Moreover, in some extreme

environments, some bacteria can interact with phagocytic cells in response to oxidative stress[58] and some can metabolize pesticides (those possessing the "Xenobiotics biodegradation and metabolism" function) or immobilize heavy metal ions to reduce their toxic effects[34,59]. Therefore, these specialized functions play important roles in ecosystems that reduce environmental stress and increase the survival probability of other microorganisms that do not have these functions. Overall, diverse specialized functional groups with irreplaceable functions in nutrient cycling and inhibitor suppression are crucial in community assembly processes. Therefore, community structures are more likely to be independent of species traits and assembly processes are more likely to be stochastic processes related to birth, death, colonization, extinction, and speciation[60].

Part II: The assembly of a low-diversity community in neutral pH soils. Soil pH can strongly influence bacterial community composition[16] and a broad range of species can grow successfully in neutral pH soils[61]; hence, the influence of abiotic habitat filtering decreases. However, diversity loss reduces specialized functions and induces a weakened metabolic network that may compromise nutrient availability and increase biotic competition. Therefore, in neutral pH soils, community assembly should be primarily driven by biological interactions (simple but strong) and selections in a heterogeneous environment (affecting only a small group of local microbes). Thus, the assembly processes are more likely to be variable selections.

Part III: The assembly of a low-diversity community in acidic and alkaline soils. When soil pH becomes increasingly extreme, an environmental filter is intensified under high or low pH conditions[16,62]. Thereby, homogeneous selection (affecting all microbes in an ecosystem simultaneously) increases without specialized functions.

Our results indicate that dilution affects the assembly processes and significantly reduces the taxonomic diversity and functional diversity of re-assembled bacterial communities. The relative abundances of specialized functional categories are decreased, whereas the relative abundances of broad functional categories are increased in low-diversity communities. This situation leads to the dominance of deterministic processes in the assembly of communities with low bacterial diversity. However, in high-diversity communities, the existence of specialized functions leads to the dominance of stochastic assembly processes. Therefore, specialized functions are potentially critical for establishing bacterial community and maintaining community functions. Further β-diversity-related investigations will improve our understanding of community assembly processes and the consequences of community functions, which will be essential to ensure the sustained provision of ecosystems.

## Methods

**Experimental design and microcosm incubation.** Black soil and red soil samples were collected from Haerbin (127°54′E, 46°28′N) in Heilongjiang Province of Northeast China and Yingtan (116°94′E, 28°21′N) in Jiangxi Province of South China, respectively. The black soil is characterized as Calcaric Chernozems and the red soil is an example of a Ferralic Cambisol according to the FAO/UNESCO System of Soil Classification. Samples were obtained from the upper 20 cm of soil perennially covered by weeds. The black soil had a pH of 8.0 and contained 4.6% organic matter, 100.8 mg kg⁻¹ N, 16.7 mg kg⁻¹ P, and 90.5 mg kg⁻¹ K. The red soil had a pH of 5.3 and contained 1.6% organic matter, 53.9 mg kg⁻¹ N, 0.95 mg kg⁻¹ P, and 61.5 mg kg⁻¹ K. Both soils were homogenized and sieved with a 2 mm sieve. A portion for preparing the inocula were temporarily stored at room temperature (20 °C) and maintained at constant moisture level (30% of field capacity) in regularly aerated bags, whereas the rest were sterilized by γ-irradiation ( > 50 kGray) (Xiyue Radiation Technology Co., Ltd, NJ, China).

Soil microcosms were prepared as described by Xun et al.[51]. Each microcosm was constructed by placing 250 g of sterilized soil into a 500 ml bottle. Sterile distilled water was added to maintain a constant moisture level of 45% of field capacity and the microcosms were pre-incubated at 20 °C in the dark for 4 weeks before sterility test on agar plates. During this 4-week pre-incubation period, lime

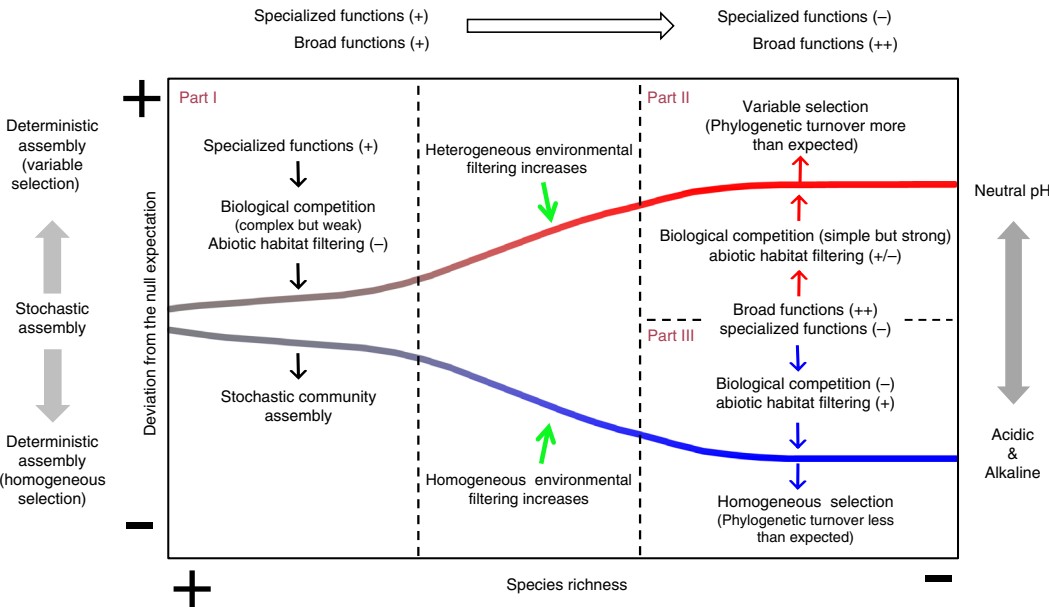

**Fig. 5** Conceptual model of bacterial community assembly processes. This conceptual model shows how diversity-related stochastic/deterministic processes contribute to the structural and functional assembly of soil bacterial communities under various pH conditions

(CaO) and ferrous sulfate (FeSO$_4$) were added (Supplementary Table 3) along with sterile distilled water to create a soil pH gradient (4.5, 5.5, 6.5, 7.5, and 8.5). The needed amount of CaO and FeSO$_4$ were obtained based on a 1-month period soil incubation pre-experiment. The $10^{-1}$ soil suspension for inoculation was made by mixing 20 g of fresh soil (10 g of black soil and 10 g of red soil) in 180 ml of sterile distilled water in a blender for 5 min. This $10^{-1}$ suspension was then progressively diluted to create the $10^{-4}$, $10^{-7}$, and $10^{-10}$ suspensions. These $10^{-1}$, $10^{-4}$, $10^{-7}$, and $10^{-10}$ suspensions were added to the microcosms of each pH level. We also included two untreated initial soils (untreated black and red soils but incubated as described for the other samples) that served as controls and all treatments were replicated six times. Therefore, we established a total of 252 microcosms [(5 pH levels × 4 dilution levels × 2 soil types + 2 initial soils) × 6 replicates]. All microcosms were then incubated at 20 °C and 45% of field capacity in the dark and agitated regularly for 16 weeks. During the entire incubation period, the bottle mouth was covered by a semi-permeable membrane to allow air exchange and bottles were only opened in a sterile biological hood. The semi-permeable membrane was replaced after each opening. Soil pH was checked every 2 weeks and the concentrations of Ca, Fe, and SO$_4{}^{2-}$ were detected using standardized method of soil physico-chemical analysis (Supplementary Table 4).

**DNA extraction and sequencing.** After 16-week incubation, two replicate samples were collected from each microcosm. Total DNA was extracted from 0.25 g of incubated soil using the PowerSoil DNA Isolation Kit (Mo Bio Laboratories, Inc., Carlsbad, CA, USA). To minimize any DNA extraction bias, three successive DNA extractions of each soil sample were pooled before performing PCR reaction. A NanoDrop ND-2000 spectrophotometer (NanoDrop, ND-2000, Thermo Scientific, 111 Wilmington, DE, USA) was used to assess DNA quality according to the 260/280 nm and 260/230 nm absorbance ratios.

Amplification of the V4 hypervariable region of the bacterial 16S rRNA gene was performed to assess the bacterial community using the primers 515 F: 5′-G TGCCAGCMGCCGCGGTAA-3′ and 806 R: 5′-GGACTACHVGGGTWTCTAA T-3′. PCR amplifications were combined in equimolar ratios and sequenced on an Illumina MiSeq instrument. The sequencing data were processed using the UPARSE pipeline (http://drive5.com/usearch/manual/uparse_pipeline.html)[63]. Raw sequences were first subjected to quality control. Singleton and chimeric sequences were removed after dereplication. The remaining sequences were categorized into operational taxonomic units (OTUs) at 97% similarity and taxonomic assignment was performed using the Silva database (Release 128) (https://www.arb-silva.de/).

The DNA extractions from red soil samples for 16S rRNA gene amplicon sequencing were also used for metagenomic DNA sequencing on an Illumina HiSeq2000 platform (150 bp paired-end reads). Low-quality reads were filtered from the Illumina raw data based upon a minimum $Q$-score of 30. All reads were assembled with IDBA 1.1.1[64] and further filtered using a minimum length of 150 bp. Finally, the assembly comprised 86,230,290 contigs, with a total length of 61.75 Gb. Genes were then predicted using MetaGeneMark v4.33[65] and clustered with a 0.95 similarity threshold using CD-HIT v4.6.2[66]. The number of reads per sample mapping to genes was calculated using SOAPaligner 2.21[67]; 39,317,785 nonredundant unigenes were obtained (Supplementary Table 5). The Kyoto Encyclopedia of Genes and Genomes (Version 58) database was used for functional gene annotation using the basic local alignment search tool.

**Statistical analysis.** For the 16S rRNA gene amplicon sequencing data, a rarefied OTU table at 11,020 reads per sample was created according to the minimum number of sequences per sample (Supplementary Table 5). Relative abundance of one phylogenetic group was defined as the number of sequences affiliated with that group divided by the total number of sequences per sample. The Shannon diversity index ($\alpha$-diversity) calculation and Bray–Curtis dissimilarity-based NMDS analysis were performed based on the rarefied OTU table. The VPA and PERMANOVA were used to determine the contributions of soil type and properties, including soil pH and the concentrations of Ca, Fe, and SO$_4{}^{2-}$, on the variation of bacterial community. The bacterial community data were normalized as percent frequency based on the rarefied OTU table and the concentrations of Ca, Fe, and SO$_4{}^{2-}$ were normalized by min–max normalization for VPA and PERMANOVA. The *bioevn* command in the vegan R package (v.2.4-1)[68] was used to select the best combination of environmental factors (best correlated with bacterial assemblage dissimilarity) and the *adonis* command in the vegan package was used to perform PERMANOVA. All the analyses above were performed using the vegan R package. The normalized percent frequency based on the rarefied OTU table was created to calculate the phylogenetic community dissimilarity ($\beta$-diversity) using FastUnifrac[69].

To infer the community assembly processes, we first calculated the mean nearest taxon distance metric using the picante R package[70] and then implemented a previously developed null modeling approach to calculate the $\beta$NTI[71,72]. $\beta$NTI > 2 is considered indicative of significantly more than expected phylogenetic turnover, which is interpreted as variable selection of deterministic processes. $\beta$NTI < −2 indicates significantly less than expected phylogenetic turnover, which is interpreted as homogeneous selection of deterministic processes. If the $|\beta\mathrm{NTI}|$ < 2, this indicates that the observed phylogenetic composition differences are the result of stochastic processes[19].

For the metagenomic DNA-sequencing data, the percent frequency of one functional category was defined as the read count abundance affiliated with that category divided by the total read count abundance per sample. The functional gene diversity was calculated using the gene richness within each functional category. The normalized metagenomic data for heatmap to demonstrate functional gene enriched or depleted pattern within each functional category was normalized by removing the mean and dividing by the SD. To determine the characteristic functional genes that were enriched or depleted by dilution, the indicator analysis (IndVal) combining both the abundance and occurrence of a given gene was used. The IndVal values were calculated using the indicspecies R package (version 1.7.6)[73]. All the correlations were calculated using Spearman's correlations. Duncan's multiple comparisons test was used to calculate the significance among samples. Tukey's honestly significant difference test was used to calculate the significance between two samples. All statistical analyses were performed using R software (version 3.3.2).

**Reporting summary.** Further information on research design is available in the Nature Research Reporting Summary linked to this article.

## Data availability
The data supporting the findings of this study are available within the paper and its Supplementary Information files. The DNA sequences from all incubation samples are deposited in the NCBI Sequence Read Archive (SRA) database with accession numbers of SRR8857587, SRR8857588, SRR8857589, SRR8857590, SRR8857591, and SRR8840928.

## Code availability
The "vegan", "picante", and "indicspecies" are packages for the R statistical language and environment. The codes for vegan (http://vegan.r-forge.r-project.org), picante (http://picante.r-forge.r-project.org), and indicspecies (http://th.archive.ubuntu.com/cran/web/packages/indicspecies/) are freely available on the web. The R codes used for calculating βNTI metrics are provided in Supplementary Note 1.

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

## Acknowledgements
We thank Professor Pascal Simonet (Universite de Lyon), Professor Shuijin Hu (North Carolina State University), and Professor George A Kowalchuk (Utrecht University) for their valuable comments on this manuscript. This research was financially supported by the National Key Basic Research Program of China (973 program, 2015CB150500), the National Natural Science Foundation for Young Scientists of China (41601252), Special and General Financial Grant from the China Postdoctoral Science Foundation (2017T100379 and 2016M601833), the Young Elite Scientists Sponsorship Program by CAST (2018QNRC001), the Science and Technology Innovation Project of Chinese Academy of Agricultural Sciences, and the Fundamental Research Funds for the Central Universities (KJQN201748).

## Author contributions
W.-B.X., R.Z., Z.X., N.Z., and Q.S. designed the study. W.L. and Y.R. performed experimental work and detailed the sampling. W.L., Y.L., and Y.M. conducted the DNA purification and organized the sequencing. W.-B.X. and W.X. carried out the bioinformatics and statistical analysis. W.-B.X. and R.Z. created the figures and drafted the manuscript. All authors helped review, edit, and complete the manuscript.

## Additional information

**Competing interests:** The authors declare no competing interests.

