## [Peer Review File · Nature Communications]

Reviewer #1 (Remarks to the Author):

The authors present an interesting investigation on links of bacterial taxonomy and function. They find a decrease in bacterial richness through dilutions that also leads to a decrease in functional gene abundances, and relative abundances of specialised functional categories are reduced in diluted communities much in line with existing literature. However, the inference that low diversity communities with deterministic community assembly features have compromised ecosystem functions and that specialised functions may be significant in community assembly and maintenance of ecosystem functions is quite novel and could be of interest to others in the community and the wider field.

The evidence for this comes mostly from analysis of relationships of β -nearest taxon index (β NTI) with dilution and pH gradients, and from shot gun metagenomics derived relative abundances of functional gene categories of “broad” and “narrow” functions. I find the observations insufficient to make the bold inferences. I am not sure how conclusive the evidence is from β NTI analysis to infer the community assembly processes. Also, the authors classification into broad and narrow functions is arbitrary (no references for the basis of such classification are given). I would consider amino acid and lipid metabolism as quite “broad” but have been classified as narrow processes. The use of relative abundances of functional genes (compositional data) also adds some limitations. When some functional genes go up, others come down. How can the authors be sure that the increase in “broad” functions in diluted communities is attributable to real shifts? On the contrary, the richness of genes in these broad categories goes down with dilutions. I think there is not enough evidence to prove that decrease in specialised functions promoted broad functions.

Some simple process rate measurements like respiration, biomass (broad functions), production of specific exo-enzymes, degradation of specific substrates (narrow functions) would have strengthened the conclusions and allowed the authors to truly link microbial function to ecosystem processes. The authors do not present the compositional taxonomic data. It may be useful to show specific taxa that survive and dominate in the diluted communities at different pH to substantiate the selection pressures either due to competition or environmental filtering.

Overall, I think the paper has the ability to contribute to the thinking in the field with some novel conclusions on a timely topical issue. But as such, with the available datasets and analysis, I see that the inferences are a bit exaggerated.

I really liked the beautiful figures. Well done!

Specific points:

Definition of specialized and broad functional categories need to be referenced or clearly justification provided

L30: devoted to specific taxa? What does that mean? Do the authors mean the specialised functions were performed by specific bacterial taxa? Taxonomic annotations of functional genes might help resolve this.

Figure 4: I am not sure if the phenomena of variable decline in functional diversity is real. The x axis representing functional gene diversity has a much smaller scale in fig 4c than that in figure 4b. Smoothing of lines connecting points in plots is also a bit misleading.

I wasn't able to find how functional gene diversity was calculated. Was it richness or beta diversity? I am assuming its richness within each category of functions.

Malik et al, 2017, mBio, volume 8, issue 4 also show soil pH related shifts in taxonomy but not functional gene diversity. They also report multiple diversity indices.

Readers would benefit from more details on statistical analysis in methods section.

There are multiple grammatical errors throughout the manuscript, I've highlighted a few below. Sentence construction is oftentimes irregular. More efforts can be put into improving the general readability of the manuscript.

Minor changes

Line 18: attention 'has' been

L21: 'differing' in

L24: 'taxonomic' diversity

L25: 'of soil bacterial communities'

L67: "A large scale investigation"? I think there are multiple biogeographical studies from different continents demonstrating soil pH as an important predictor of bacterial diversity.

L170: 'magnitude level' redundant words

L290: Unesco is capitalized

L 131: "we detected"

L131: 'shift patterns' redundant words

L139-141: consider revising

L194: stepwise diversity descent patterns? Very confusing phrase

L207: grammar correction required

L222: 'effect of pH increased' not decreased!!

Reviewer #2 (Remarks to the Author):

Weibing Xun and collaborators have studied the assembly of bacterial community, based on 16S rRNA partial gene amplicon, in 2 very different gamma sterilized soils (black and red soils), inoculated with different dilutions of the original soils. In addition, these 2 soils were subjected to different levels of chemical disturbances with CaO and FeSO₄, what changes the soil pH to different levels (4.5, 5.5, 6.5, 7.5 and 8.5). The results showed that the alpha diversity of high-diluted soils were lower than those of less diluted, what is not novel. In order to know what are the potential functions, based on DNA shotgun, changed in different diluted soils and link them to alpha diversity, the authors sequenced the DNA of the treatments of one soil, the Black soil. The authors concluded that the "relative abundances of specialized function categories were lower, while the relative abundances of broad function categories were higher in diluted communities. Consequently, assembly processes significantly differed among dilution levels; Overall, they suggested that the low-diversity-related deterministic community assembly processes compromise ecosystem functions, highlighting the crucial roles of specialized functions in community assembly and in generating and maintaining soil ecosystem functions."

The scientific question of this study is very relevant and with the dataset obtained, the authors may have the responses to the question. However, I have major concerns about how the data was analyzed and more importantly, I missed the biological and ecological explanations on the bacterial groups/communities found in each treatment and their link with their potential functions in soil ecosystem functioning. Here are a list of few major concerns:

The 2 soils were very different in pH and nutrient contents. Black soil was extremely rich in nutrients as compared to Red soil. The Black soil has pH 8.0 while Red soil has pH 5.3. Beside high nutrient contents, Black soil has pH above neutral, what condition makes the nutrients available to plants and microbes while the Red soil is the opposite.

Changing of soil pH with CaO (increase pH) or with FeSO₄ (decrease pH) affects tremendously the soil chemistry and in addition, changes the chemical composition of the soil. Black soil had addition of Fe and SO₄ and Red soil had the addition of CaO. These elements are disturbances that will affect the bacterial assemblage, thus the differences in bacterial community is not because of the pH but because of the elements added into soil. It is well known that elements Ca, Fe and SO₄ affected bacteria. The authors did not do any statistically analyses to determine what were the main factors

that explain such differences (they could use PERMANOVA). Although they stated in lines 104-106 “Moreover, although different amounts of CaO and FeSO₄ were used to manipulate soil pH, the concentrations of Ca and Fe had little impact on the reassembled communities.” To state this they should have determined the concentrations of added Ca, Fe and SO₄ and the initial contents of these elements in the original Black and Red soils and statistically showed the differences.

Why focus on FunGp1 and FunGp2 categories in this experiment? If we look at the 2 soil chemical compositions, there is a huge difference between the 2 soils on organic matter, nitrogen and phosphorus contents. The C, N and P cycles are essential cycles in soil functioning. So, why not to focus on these cycles? In addition, because the authors added Sulphur (FeSO₄) to the Black soil to decrease the pH, why not to focus in this cycle as well?

There are missing many details on MM and on how the data was analyzed, specially the statistical methods. Examples are:

L 340-341: how the OTU were rarefied? How the data was normalized for beta diversity analyses?

How the shotgun data was normalized?

What was the statistical method used to infer that the functional categories were different?

What kind of data (abundances, diversity, etc...) were used to determine the figures 2c, 2d, 2e, 2f, 2g, 2h? How the correlations were calculated?

How the needed amount of CaO and FeSO₄ to reach desired pH was calculated and checked?

L334- Where are the number of reads/ per sample? No table.

Responses to reviewer #1:

General comments:

The authors present an interesting investigation on links of bacterial taxonomy and function. They find a decrease in bacterial richness through dilutions that also leads to a decrease in functional gene abundances, and relative abundances of specialized functional categories are reduced in diluted communities much in line with existing literature. However, the inference that low diversity communities with deterministic community assembly features have compromised ecosystem functions and that specialized functions may be significant in community assembly and maintenance of ecosystem functions is quite novel and could be of interest to others in the community and the wider field.

Q1-1: The evidence for this comes mostly from analysis of relationships of β -nearest taxon index (β NTI) with dilution and pH gradients, and from shot gun metagenomics derived relative abundances

of functional gene categories of “broad” and “narrow” functions. I find the observations insufficient to make the bold inferences. I am not sure how conclusive the evidence is from β NTI analysis to infer the community assembly processes.

Response: Thanks for the recognition of the novelty of our study. In this study, the definition of community assembly processes (β NTI analysis) is an important part relating to dilution (α -diversity) and functional groups (broad and specialized functions).

For community assembly, historically, ecologists have debated along two distinct theoretical lines to examine and interpret community assembly: the niche-based theory and the neutral theory. Nowadays, this polarized dichotomy has been suppressed, as it has become more broadly accepted that both stochastic and deterministic processes influence community assembly simultaneously.

To infer on the influences of ecological processes structuring community assembly, Webb et al. (2002) laid out a heuristic framework to examine deviations between phylogenetic community structure and the null expectation by randomization procedures, referred to as the “phylogenetic null model”. This framework is based on the assumption that closely related species have similar ecological niches, and that species coexistence patterns depend on competition or physiological tolerance. An extended method, which combines phylogenetic community structure and standardized Bray–Curtis matrices with null models, builds on Webb’s framework by allowing the identification of the ecological processes mediating community assembly. The use of phylogenetic information enables the elucidation of the phylogenetic relatedness of species within and across communities. Between-community mean nearest taxon distance (β MNTD), which is based on the mean distance to the nearest neighbor, is a measure of the clustering of closely related species. Null model distributions of β MNTD can be built by shuffling species among the tips of the phylogenetic tree or between different communities using

randomizations and permutation analyses. The β -nearest taxon index (β NTI) is then generated by comparing the observed and the null distribution of β MNTD. Within this model, ecological processes are divided into deterministic (variable selection and homogeneous selection) and stochastic processes. Variable selection results in high compositional turnover, where environmental factors (biotic and abiotic) shift temporally or spatially. In contrast, homogeneous selection leads to low compositional turnover, where the selection is consistent through time and space. Stochastic process indicates that the observed difference in phylogenetic community composition is not the result of deterministic processes. It should be the result of dispersal limitation (very low dispersal rates), homogenizing dispersal (very high dispersal rates), or undominated process (dispersal and drift contribute equally to community composition). This β NTI analysis for estimating community assembly processes has also been used in a number of studies, i.e. Dini-Andreote et al, 2015, PNAS; Stegen et al, 2016, Nature Communications; Tripathi et al, 2018, ISME J. Therefore, we consider that the β NTI analysis based on community turnover is suitable to identify the community assembly processes in our experiment. And also, we suggest that the dilution-induced diversity loss has great impacts on the functional traits and community assembly processes.

Q1-2: Also, the authors' classification into broad and narrow functions is arbitrary (no references for the basis of such classification are given). I would consider amino acid and lipid metabolism as quite "broad" but have been classified as narrow processes.

Response: This is a very important query. The definition of broad and specialized functional categories is quite necessary to eliminate the confusion of readers if published. A total of 17 metabolism-related functional categories were detected in this study. 10 out of these 17 functional gene categories were

defined as broad functions while the rest were defined as specialized functions. In this revision, we provide a detailed description of the functional classification (detailed in the “Definition of broad and specialized functional categories” section in Methods part). **(L349-376 in the revised manuscript)**

The “Amino acid metabolism” and “Lipid metabolism” functions are quite widely distributed. For instance, the biosynthesis of iso- and anteiso-fatty acids occur in many bacteria as the major acyl constituents of membrane lipids and the biosynthesis pathways of non-essential amino acid are distributed broadly among diverse bacteria. However, the biosynthesis of omega-cyclohexyl and omega-cycloheptyl fatty acids and the metabolisms of some essential amino acids are only present in several bacterial species. Although amino acids and lipid are small molecule compounds and a part of them are really widely distributed, we know quite a part are still relatively narrow distributed. Therefore, the “Amino acid metabolism” and “Lipid metabolism” functions were defined as specialized functions in this study.

We acknowledge this classification may cause some controversy, since these two functional categories seem to be situated between broad and specialized functions. However, according to the information we gathered from previous researches and the results we got from our study (showed in Figure 4a that the “Amino acid metabolism” and “Lipid metabolism” functional categories were detected in higher relative abundances in less diluted samples than in more diluted samples), we think it is better to define these two functional categories as specialized functions.

Q1-3: The use of relative abundances of functional genes (compositional data) also adds some limitations. When some functional genes go up, others come down. How can the authors be sure that the increase in “broad” functions in diluted communities is attributable to real shifts? On the contrary,

the richness of genes in these broad categories goes down with dilutions. I think there is not enough evidence to prove that decrease in specialized functions promoted broad functions.

Response: The assessment whether the functional genes go up or come down were all based on the relative abundances. We remove the specialized functional species (rare species) from the community; a greater proportion of reads will be sequenced as broad functions. So the broad functions did not necessarily go up in abundance but the relative abundance. Meanwhile, the term “promoted” in the initial manuscript may go to be misleading, so we used a conciliatory description (“Therefore, we suggested that the decrease of specialized functions synchronously resulted in a greater proportion of broad functions”). **(L207-209 in the revised manuscript)**

Q1-4: Some simple process rate measurements like respiration, biomass (broad functions), production of specific exo-enzymes, degradation of specific substrates (narrow functions) would have strengthened the conclusions and allowed the authors to truly link microbial function to ecosystem processes. The authors do not present the compositional taxonomic data. It may be useful to show specific taxa that survive and dominate in the diluted communities at different pH to substantiate the selection pressures either due to competition or environmental filtering.

Overall, I think the paper has the ability to contribute to the thinking in the field with some novel conclusions on a timely topical issue. But as such, with the available datasets and analysis, I see that the inferences are a bit exaggerated.

I really liked the beautiful figures. Well done!

Response: Thank you for your comments. Actually, some similar explorations have been addressed to evaluate the linkage between biodiversity and function. For instance, soil respiration (broad function,

Griffiths et al. 2001) shows no differences while potential denitrification activity (specialized function, Philippot et al. 2013) and pesticide mineralization capacity (specialized function, Singh et al. 2014) decrease with soil bacterial diversity loss. In our study, although we did not do the measurement like respiration, biomass or production of specific enzymes, we found very attractive results on the metagenomic data. These results provide a basic assessment of soil potential functions that allow us to associate these functions with microbial diversity and community assembly processes. Moreover, we have noticed that the compositional data of the re-assembled communities contains some important information. In this revised manuscript, we have provided some compositional analyses of these communities and added some descriptions on these results. **(L79-84 in the revised manuscript)**

Specific comments:

Q1-5: Definition of specialized and broad functional categories needs to be referenced or clearly justification provided.

Response: It's a very valuable comment. As we have responded in the previous query (Q1-2 of the General comments). The definition of broad and specialized functional categories is quite necessary to eliminate the confusion of readers if published. A total of 17 metabolism-related functional categories were detected in this study. 10 out of these 17 functional gene categories were defined as broad functions while the rest were defined as specialized functions. In this revision, we provide a detailed description of the functional classification (detailed in the "Definition of broad and specialized functional categories" section in Methods part). **(L349-376 in the revised manuscript)**

Q1-6: L30: devoted to specific taxa? What does that mean? Do the authors mean the specialized functions were performed by specific bacterial taxa? Taxonomic annotations of functional genes might help resolve this.

Response: That's correct. The specialized functions are not so widely distributed that should be performed by narrow groups of bacterial taxa. Thanks for your comment on the compositional analysis. Taxonomic annotations could potentially provide evidence for specific bacterial taxa. Therefore, we provide compositional analyses of these communities and added a brief description on these results in the revised manuscript. **(L79-84 in the revised manuscript)**

Q1-7: Figure 4: I am not sure if the phenomenon of variable decline in functional diversity is real. The x axis representing functional gene diversity has a much smaller scale in fig 4c than that in figure 4b. Smoothing of lines connecting points in plots is also a bit misleading.

Response: The phenomenon of variable decline in functional diversity is the smoothing fitting curve that connecting functional gene diversity points under increasing dilution levels. We used these smooth curves to exhibit how the functional gene diversity responds to dilution. Here, we fitted these curves using the same scale in x axis again and found the same phenomenon (showed in the figure below). In addition, the main point exhibited by these curves was that the functional gene diversity of FunGp1 and FunGp2 were both declined with increasing dilution levels and the functional diversity decrease amplitudes of FunGp1 were greater than FunGp2. However, these smoothing fitting curves of two functional groups showed different phenomenon, which we do not know if it is real, thus may exist some misleading as you are concerned. So we removed the comparison of the variable decline

phenomenon between FunGp1 and FunGp2, and emphasized the decreasing diversity and greater decrease amplitudes in FunGp1. **(L155-158 & 196-200 in the revised manuscript)**

Q1-8: I wasn't able to find how functional gene diversity was calculated. Was it richness or beta diversity? I am assuming its richness within each category of functions.

Response: Sorry for the unclear description. Yes, the functional gene diversity was calculated using the gene richness within each functional category rather than the beta diversity. The relative abundance of one functional category was defined as the number of gene reads affiliated with that category divided by the total number of gene reads per sample. And then, the functional gene diversity was calculated using the gene richness within each functional category. The detailed description has been added to the Methods section of the revised manuscript. **(L400-403 in the revised manuscript)**

Q1-9: Malik et al, 2017, mBio, volume 8, issue 4 also show soil pH related shifts in taxonomy but not functional gene diversity. They also report multiple diversity indices.

Response: This is also a key point that we found although the bacterial composition were different across different pH levels, the relative abundances of every functional category were less significantly

differed, or in some cases not at all. Thanks for pointing out. This point has been discussed and this reference has been added to the revised manuscript. **(L226-227 in the revised manuscript)**

Q1-10: Readers would benefit from more details on statistical analysis in methods section.

Response: The detailed description of statistical analysis has been provided in the Methods section.

Changes are listed below:

(1) How to rarefy OTU table: “a rarefied OTU table at 11020 reads per sample was created according to the minimum number of sequences per sample.” **(L378-379 in the revised manuscript)**

(2) How to conduct the VPA and PERMANOVA, including the R packages and commands: “The Variation partitioning analysis (VPA) and permutational multivariate analysis of variance (PERMANOVA) were used to determine the contributions of soil type and properties, including soil pH and the concentrations of Ca, Fe and SO_4^{2-} , on the variation of bacterial community. All the analyses above were performed using the vegan R package (v.2.4-1). The bioenv command in the vegan package was used to select the best combination of environmental factors (best correlated with bacterial assemblage dissimilarity) and the adonis command in the vegan package was used to perform PERMANOVA.” **(L383-390 in the revised manuscript)**

(3) How the data was normalized for β -diversity analyses: “The normalized percent frequency based on the rarefied OTU table was created to calculate the phylogenetic community dissimilarity (β -diversity) using FastUnifrac” **(L391-393 in the revised manuscript)**

(4) How to normalize the metagenomic data and how to calculate the diversity of each functional categories: “For the metagenomic DNA sequencing data, the percent frequency of one functional category was set as the read count abundance affiliated with that category divided by the total read

count abundance per sample. The functional gene diversity was calculated using the gene richness within each functional category. The normalized metagenomic data for heatmap to demonstrate functional gene enriched or depleted pattern within each functional category was normalized by removing the mean and dividing by the standard deviation.” (L400-405 in the revised manuscript)

Q1-11: There are multiple grammatical errors throughout the manuscript. I’ve highlighted a few below. Sentence construction is oftentimes irregular. More efforts can be put into improving the general readability of the manuscript.

Minor changes

Line 18: attention ‘has’ been

L21: ‘differing’ in

L24: ‘taxonomic’ diversity

L25: ‘of soil bacterial communities’

L67: “A large scale investigation”? I think there are multiple biogeographical studies from different continents demonstrating soil pH as an important predictor of bacterial diversity.

L170: ‘magnitude level’ redundant words

L290: Unesco is capitalized

L 131: “we detected”

L131: ‘shift patterns’ redundant words

L139-141: consider revising

L194: stepwise diversity descent patterns? Very confusing phrase

L207: grammar correction required

L222: 'effect of pH increased' not decreased!!

Response: Thank you for these language editing suggestions. All of the highlighted grammatical errors have been addressed. Moreover, to improve the readability of this manuscript, the revised manuscript has been improved and grammatically checked by two English native speakers. We hope this revision will meet the requirements of Nature Communications. All changes made in the revised manuscript were highlighted with colored text.

Responses to reviewer #2:

General comments:

Weibing Xun and collaborators have studied the assembly of bacterial community, based on 16S rRNA partial gene amplicon, in 2 very different gamma-sterilized soils (black and red soils), and inoculated with different dilutions of the original soils. In addition, these 2 soils were subjected to different levels of chemical disturbances with CaO and FeSO₄, what changes the soil pH to different levels (4.5, 5.5, 6.5, 7.5 and 8.5). The results showed that the alpha diversity of high-diluted soils were lower than those of less diluted, what is not novel. In order to know what are the potential functions, based on DNA shotgun, changed in different diluted soils and link them to alpha diversity, the authors sequenced the DNA of the treatments of one soil, the Black soil. The authors concluded that the “relative abundances of specialized function categories were lower, while the relative abundances of broad function categories were higher in diluted communities. Consequently, assembly processes significantly differed among dilution levels; overall, they suggested that the low-diversity-related deterministic community assembly processes compromise ecosystem functions, highlighting the crucial roles of specialized functions in community assembly and in generating and maintaining soil ecosystem functions.”

Q2-1: The scientific question of this study is very relevant and with the dataset obtained, the authors may have the responses to the question. However, I have major concerns about how the data was analyzed and more importantly, I missed the biological and ecological explanations on the bacterial groups/communities found in each treatment and their link with their potential functions in soil ecosystem functioning.

Response: Thanks for your valuable comments.

1. The detailed description of statistical analysis has been provided in the Methods section.

Changes are listed below:

(1) How to rarefy OUT table: “a rarefied OTU table at 11020 reads per sample was created according to the minimum number of sequences per sample.” **(L378-379 in the revised manuscript)**

(2) How to conduct the VPA and PERMANOVA, including the R packages and commands: “The Variation partitioning analysis (VPA) and permutational multivariate analysis of variance (PERMANOVA) were used to determine the contributions of soil type and properties, including soil pH and the concentrations of Ca, Fe and SO₄²⁻, on the variation of bacterial community. All the analyses above were performed using the vegan R package (v.2.4-1). The bioenv command in the vegan package was used to select the best combination of environmental factors (best correlated with bacterial assemblage dissimilarity) and the adonis command in the vegan package was used to perform PERMANOVA.” **(L383-390 in the revised manuscript)**

(3) How the data was normalized for β -diversity analyses: “The normalized percent frequency based on the rarefied OTU table was created to calculate the phylogenetic community dissimilarity (β -diversity) using FastUnifrac” **(L391-393 in the revised manuscript)**

(4) How to normalize the metagenomic data and how to calculate the diversity of each functional categories: “For the metagenomic DNA sequencing data, the percent frequency of one functional category was set as the read count abundance affiliated with that category divided by the total read count abundance per sample. The functional gene diversity was calculated using the gene richness within each functional category. The normalized metagenomic data for heatmap to demonstrate functional gene enriched or depleted pattern within each functional category was normalized by removing the mean and dividing by the standard deviation.” **(L400-405 in the revised manuscript)**

2. We realized that the compositional data of these re-assembled communities contain important information. In this revised manuscript, we have provided some compositional analyses of these communities. **(L79-84 in the revised manuscript)**

Q2-2: The 2 soils were very different in pH and nutrient contents. Black soil was extremely rich in nutrients as compared to Red soil. The Black soil has pH 8.0 while Red soil has pH 5.3. Beside high nutrient contents, Black soil has pH above neutral, what condition makes the nutrients available to plants and microbes while the Red soil is the opposite.

Response: Thank you for your comments. Yes, these two soils were very different regarding pH and nutrient contents. We used these two distinct soils in our research to test if the communities show similar assembly processes on soil pH and dilution gradients, which will make our conclusion more general and solid.

Q2-3: Changing of soil pH with CaO (increase pH) or with FeSO₄ (decrease pH) affects tremendously the soil chemistry and in addition, changes the chemical composition of the soil. Black soil had

addition of Fe and SO₄ and Red soil had the addition of CaO. These elements are disturbances that will affect the bacterial assemblage, thus the differences in bacterial community is not because of the pH but because of the elements added into soil. It is well known that elements Ca, Fe and SO₄ affected bacteria. The authors did not do any statistically analyses to determine what were the main factors that explain such differences (they could use PERMANOVA). Although they stated in lines 104-106 “Moreover, although different amounts of CaO and FeSO₄ were used to manipulate soil pH, the concentrations of Ca and Fe had little impact on the reassembled communities.” To state this they should have determined the concentrations of added Ca, Fe and SO₄ and the initial contents of these elements in the original Black and Red soils as statistically showed the differences.

Response: Thank you for the valuable suggestion. We measured the concentrations of Ca, Fe and SO₄²⁻ in all soil samples (**this data has been provided as supplementary table 2**). The *bioenv* command in the vegan package of R was used to select the best combination of environmental factors and the concentrations of Ca and Fe were selected for Variation partitioning analysis (VPA) (**Detailed description for VPA in Statistical analysis section. L383-390 in the revised manuscript**). We found they had little impact on soil bacterial community variations. However, we did not know the separate contributions of Ca, Fe and SO₄²⁻ on the variation of bacterial communities. Therefore, we conducted the PERMANOVA as you suggested. The results demonstrated that the concentrations of Ca, Fe and SO₄²⁻ explained only 3.0% (PERMANOVA, $F_{9, 470} = 1.97$, $p = 0.037$), 2.4% (PERMANOVA, $F_{9, 470} = 1.49$, $p = 0.063$) and 1.1% (PERMANOVA, $F_{9, 470} = 1.90$, $p = 0.046$) of the variances in bacterial community composition, respectively. (**L110-116 in the revised manuscript**)

Q2-4: Why focus on FunGp1 and FunGp2 categories in this experiment? If we look at the 2 soil chemical compositions, there is a huge difference between the 2 soils on organic matter, nitrogen and phosphorus contents. The C, N and P cycles are essential cycles in soil functioning. So, why not to focus on these cycles? In addition, because the authors added Sulphur (FeSO_4) to the Black soil to decrease the pH, why not to focus in this cycle as well?

Response: In this study, we are trying to assess how dilution and soil pH impact soil potential functions of the re-assembled bacterial communities. Since we detected similar shift patterns in community assembly processes and bacterial diversity in response to soil pH and dilution, the shotgun metagenomic analyses were conducted only based on the DNA extractions from the red soil samples. Therefore, we were not going to compare the functions between black soil and red soil. Additionally, these two groups of functional gene categories we analyzed in this study also include the categories of nitrogen, sulfur and carbohydrate metabolisms.

Q2-5: There are missing many details on MM and on how the data was analyzed, specially the statistical methods. Examples are:

L 340-341: how the OTU were rarefied? How the data was normalized for beta diversity analyses?

How the shotgun data was normalized?

What was the statistical method used to infer that the functional categories were different?

What kind of data (abundances, diversity, etc...) were used to determine the figures 2c,2d, 2e, 2f, 2g,

2h? How the correlations were calculated?

How the needed amount of CaO and FeSO_4 to reach desired pH was calculated and checked?

L334- Where are the number of reads/ per sample? No table.

Response: Thanks for your suggestion. The detailed descriptions of statistical analysis and more about the experimental design have been provided in the Methods section as you suggested.

Changes are listed below (The repetitive changes to the response in **Q2-1** are only listed here):

- (1) How to rarefy OUT table. **(L378-379 in the revised manuscript)**
- (2) How to conduct the VPA and PERMANOVA, including the R packages and commands. **(L383-390 in the revised manuscript)**
- (3) How the data was normalized for β -diversity analyses. **(L391-393 in the revised manuscript)**
- (4) How to normalize the metagenomic data and how to calculate the diversity of each functional categories. **(L400-405 in the revised manuscript)**
- (5) How to define functional categories.

The definition of broad and specialized functional categories is quite necessary to eliminate the confusion of readers if published. A total of 17 metabolism-related functional categories were detected in this study. 10 out of these 17 functional gene categories were defined as broad functions while the rest were defined as specialized functions. In this revision, we provide a detailed description of the functional classification (detailed in the “Definition of broad and specialized functional categories” section in Methods part). **(L349-376 in the revised manuscript)**

- (6) What data were used to determine the figures 2? How the correlations were calculated?

The bacterial community data were normalized as percent frequency based on the rarefied OTU table and the concentrations of Ca, Fe and SO_4^{2-} were normalized by min-max normalization for VPA and PERMANOVA **(L386-388 in the revised manuscript)**. All the correlations were calculated using Spearman correlations **(L393 in the revised manuscript)**.

- (7) How the needed amount of CaO and FeSO_4 to reach desired pH was calculated and checked?

The needed amount of CaO and FeSO₄ were obtained based on a one-month period soil incubation pre-experiment (**The amount of CaO and FeSO₄ we added to manipulate soil pH was provided in Supplementary Table 1**). Soil pH was checked every two weeks and the concentrations of Ca, Fe and SO₄²⁻ were detected using standardized method of soil physic-chemical analysis. (**L307-308 & 319-321 in the revised manuscript**)

(8) The number of reads/ per sample?

To provide more detailed information on our sequencing data. The number of contigs after assembly, the total length of contigs, the number of nonredundant unigenes from the metagenomic sequencing data, and the minimum number of reads per sample we obtained from the amplicon sequencing data have all been provided in the revised manuscript (**Supplementary Table 3**).

Reviewer #1 (Remarks to the Author):

The authors have quickly responded to reviewer suggestions and some of the changes have improved the manuscript. For example, more details of the statistical steps in data analysis are now presented. I still have some concerns and they are as follows:

The authors' response about the classification of functions into broad and narrow is not satisfactory. I suggest changes in the nomenclature here which actually does not change the main results that certain specialised functions get lost due to dilutions. The authors argue that part of genes belonging to "Amino acid metabolism" and "Lipid metabolism" functions are widely distributed but a part is also narrowly distributed. Yet choose to call them specialised BECAUSE they were detected in lower abundance in less diluted sample, something they set out to test in the first place. Sounds like its classified so to fit the trend.

These functional classes need not be called specialised or broad, they could be in-between. Even if they call them broad functions that class could behave differently from other classes of broad functions. Biology is complex and trends need not be coherent. The results don't change because the clearly identifiable specialised functions like sulphur, nitrogen, methane, xenobiotics metabolism are lost on dilution. This is the key result and holds good regardless.

A general problem with using upper level functions in omics datasets is that the functional groups at this level consist of a myriad of genes that may decrease or increase in response to environmental stimuli/perturbations. Often multiple genes may respond to move in opposite directions and yet at upper level of functional groups they may cancel out and give an impression that the entire functional group has not responded to that particular treatment. Therefore, it is problematic to make conclusions solely based on shifts in upper level functional groups without considering the shifts within those groups.

A solution could perhaps be analysing the data at the gene level. In addition to doing the analysis at this upper level of functional categories the same could be applied to individual genes/functions. Of course, there will be thousands of genes but we now have the capabilities to do such analysis. The authors could look at the direction of change on dilution for each gene and quantify the frequency of gens that increase, decrease or don't change within each upper level functional groups. An approach of looking at metagenomics data at various levels of hierarchical classification could be very useful. For pairwise data indicator analysis (indval) can be used to pick out genes enriched in either groups. As a reductionist approach, the authors could apply a pairwise analysis to the undiluted and most diluted samples.

With this analysis, they could also highlight some of the specific genes within the specialised categories of sulphur, nitrogen, methane, xenobiotics metabolism, etc that are lost on dilution and

what is the functional relevance of loss of those specific genes over others. It will only give strength to the conclusions.

The taxonomic data presented now in the revised version is very interesting. It looks like certain groups (Acido and Actinobacteria) get selectively lost during dilutions. This is a very important observation and could actually explain the loss of certain specialised functions. Yet, there is no discussion of these results in the MS.

I disagree with reviewer 2 that the functional assessment should be limited to genes involved in C, N, P cycles and endorse the approach taken by the authors to analyse all functional groups. Because, microbial physiology and fitness in its entirety including housekeeping metabolic pathways dictates how communities affect ecosystem function.

Reviewer #2 (Remarks to the Author):

My main concern of this study is the design of the experiment that led to the conclusion.

The conclusion of this study is based on a single soil type (red soil). Statistically, it is not possible to conclude what is in the title 'Diversity-triggered deterministic microbiome assembly constrains ecosystem functions', which is misleading. To conclude that would be necessary at least 3 different soils.

This study is a description of what happens with the regrowth of bacterial community composition (in red and black soils) and potential function in one specific soil (red soil) when they are diluted and inoculated in the origin sterilized soils. The result as indicated by the authors "the relative abundances of specialized functional categories were lower, while the relative abundances of broad functional categories were higher in more diluted communities." is expected because the microbes that grew in diluted soils were fast growing (copiotrophs) microbes (Betaproteobacteria, Gammaproteobacteria and Bacteroidetes), due to high nutrient condition of the soil.

This study is interesting but lacks novelty to be published in Nature communications. I would suggest to be published in a different journal.

Responses to reviewer #1:

Q1-1: The authors have quickly responded to reviewer suggestions and some of the changes have improved the manuscript. For example, more details of the statistical steps in data analysis are now presented. I still have some concerns and they are as follows:

The authors' response about the classification of functions into broad and narrow is not satisfactory.

I suggest changes in the nomenclature here which actually does not change the main results that certain specialized functions get lost due to dilutions. The authors argue that part of genes belonging to "Amino acid metabolism" and "Lipid metabolism" functions are widely distributed but a part is also narrowly distributed. Yet choose to call them specialized BECAUSE they were detected in lower abundance in less diluted sample, something they set out to test in the first place. Sounds like its classified so to fit the trend.

These functional classes need not be called specialized or broad, they could be in-between. Even if they call them broad functions that class could behave differently from other classes of broad functions. Biology is complex and trends need not be coherent. The results don't change because the clearly identifiable specialized functions like sulfur, nitrogen, methane, xenobiotics metabolism are lost on dilution. This is the key result and holds good regardless.

Response: Thanks for your suggestions on the classification of functions. We agree with your opinion that "Amino acid metabolism" and "Lipid metabolism" are generally belong to broad functions, but not all these gene categories in the broad functions follow the depletion trend (**L170-173 of the**

Revised

Manuscript). We also provided the detailed analysis on gene level as suggested, and then discussed this inconsistent situation and tried to explain this in the discussion **(L226-233 of the Revised Manuscript).**

Q1-2: A general problem with using upper level functions in omics datasets is that the functional groups at this level consist of a myriad of genes that may decrease or increase in response to environmental stimuli/perturbations. Often multiple genes may respond to move in opposite directions and yet at upper level of functional groups they may cancel out and give an impression that the entire functional group has not responded to that particular treatment. Therefore, it is problematic to make conclusions solely based on shifts in upper level functional groups without considering the shifts within those groups.

A solution could perhaps be analyzing the data at the gene level. In addition to doing the analysis at this upper level of functional categories the same could be applied to individual genes/functions. Of course, there will be thousands of genes but we now have the capabilities to do such analysis. The authors could look at the direction of change on dilution for each gene and quantify the frequency of genes that increase, decrease or don't change within each upper level functional group. An approach of looking at metagenomics data at various levels of hierarchical classification could be very useful. For pairwise data indicator analysis (indval) can be used to pick out genes enriched in either groups. As a reductionist approach, the authors could apply a pairwise analysis to the undiluted and most diluted samples.

With this analysis, they could also highlight some of the specific genes within the specialized categories of sulfur, nitrogen, methane, xenobiotics metabolism, etc that are lost on dilution and what is

the functional relevance of loss of those specific genes over others. It will only give strength to the conclusions.

Response: Thanks for these valuable suggestions. The additional gene-level analyses was really helpful to consolidate the conclusions of this study. Followed your instruction, we analyzed all of unique genes within each functional category. As you predicted, in every functional category of both specialized (FunGp1) and broad (FunGp2) groups, along the dilution level, all three situations of increasing, decreasing and no changing of the relative abundance were found, but with different proportions. The general trend that “The relative abundances of specialized functional categories are decreased, while the relative abundances of broad functional categories are increased in low-diversity communities” was still existed. For FunGp1, large proportions (31.45% ~ 64.87%) of the genes within these categories were significantly decreased along the dilution level, whereas only small proportions (1.67% ~ 9.22%) were significantly increased; while for FunGp2, large proportions (34.89% ~ 61.95%) of the genes were significantly increased with dilution level, while only small proportions (1.98% ~ 8.12%) of these genes were significantly decreased. We added a supplementary table (Table S1) to summarize the frequency of increased and decreased genes within each category along the dilution in the revised manuscript. Also, as suggested, we did the indicator analysis to pick up some indicator genes (listed in supplementary table 2) and discussed their relevance as well as the possible link with microbial species lost. We think your valuable suggestions and these additional analyses made the conclusions more strong (**L153-161 & L165-173 of the Revised Manuscript**). The statistical method of the indicator analysis has been provided in the Methods section (**L415-418 of the Revised Manuscript**).

Q1-3: The taxonomic data presented now in the revised version is very interesting. It looks like certain groups (Acido and Actinobacteria) get selectively lost during dilutions. This is a very important observation and could actually explain the loss of certain specialized functions. Yet, there is no discussion of these results in the MS.

I disagree with reviewer 2 that the functional assessment should be limited to genes involved in C, N, P cycles and endorse the approach taken by the authors to analyze all functional groups. Because, microbial physiology and fitness in its entirety including housekeeping metabolic pathways dictates how communities affect ecosystem function.

Response: Thank you for these comments. The discussion of taxonomic results has now been added to the revised manuscript. “Numerous studies suggested that some bacterial groups of *Acidobacteria*, *Actinobacteria* and *Nitrospira* are keystone taxa which may harbour specialized functions like nitrogen fixation or ammonia oxidation. For instance, the gene clusters inferring to “Terpenoids and Polyketides metabolism” usually harboured by a minority of soil microbial community like some groups of *Actinobacteria*. Several investigations have demonstrated that the enzymes involved in “Xenobiotics biodegradation/metabolism” are inducible, and only exist in a small group of microorganisms. Consistently, the taxonomic composition results in our study demonstrated that the bacterial groups of *Acidobacteria*, *Actinobacteria* and *Nitrospira* were depleted in more diluted samples. Thus, the dilution-induced loss of rare species may significantly constrain the physiological pathways of specialized functions and result in a weakened metabolic network.” (L213-221 of the Revised

Manuscript)

Responses to reviewer #2:

Q2-1: My main concern of this study is the design of the experiment that led to the conclusion.

The conclusion of this study is based on a single soil type (red soil). Statistically, it is not possible to conclude what is in the title 'Diversity-triggered deterministic microbiome assembly constrains ecosystem functions', which is misleading. To conclude that would be necessary at least 3 different soils.

This study is a description of what happens with the regrowth of bacterial community composition (in red and black soils) and potential function in one specific soil (red soil) when they are diluted and inoculated in the origin sterilized soils. The result as indicated by the authors "the relative abundances of specialized functional categories were lower, while the relative abundances of broad functional categories were higher in more diluted communities." is expected because the microbes that grew in diluted soils were fast growing (copiotrophs) microbes (Betaproteobacteria, Gammaproteobacteria and Bacteroidetes), due to high nutrient condition of the soil.

This study is interesting but lacks novelty to be published in Nature communications. I would suggest to be published in a different journal.

Response: Thank you for your review and comments on this manuscript. In this study, we try to link the functional traits to the community assembly and ecosystem functioning. We found that stochastic assembly processes were dominant in high-diversity communities. However, deterministic assembly processes were dominant in low-diversity communities due to the reduction of specialized functions devoted to specific bacterial taxa. Overall, we suggest that the low-diversity-induced deterministic community assembly processes constrain ecosystem functions, highlighting the crucial roles of specialized functions in community assembly and in generating and sustaining the function of soil ecosystems. We consider that the topic of diversity loss-triggered depletion of specialized functions

may be significant in community assembly and ecosystem functions maintenance is quite novel.

Reviewer #1 (Remarks to the Author):

Reviewer recommendations have been implemented by the authors and the results of additional analysis seem to be in line with the presented results. I am content but still not excited with the manuscript in its current state.

Additional comments:

No ecosystem function measures were made. Title, abstract and the manuscript text should refer to community function.

Also recommend replacing microbiome with bacteria in the title.

Line 29-31: These linkages are only correlative. Functional genes were not annotated to taxonomy, the authors should highlight that these are only correlations. Devoted is not the best term to use here.

Line 31-33: Finding a functional gene in the community does not indicate ecosystem consequences. A metagenome provides a snapshot of potential function of the community. Better use the term community function or such. Again, these are all correlative linkages so recommend using words like “probably” or “likely” so that readers are not misled.

Line 147: carried out

Line 154, 156, 165, 167: Why ~ to show range?

Line 206: on dilution

Line 213: have suggested

Line 216: are usually

Line 228: occurs

Line 318: Taxonomic diversity?

Line 323: At what point?

Line 324: soil community potential functions were assessed in this study

There are still a lot of typos and grammatical errors, I've highlighted some here but I recommend careful review for language errors.

Reviewer #3 (Remarks to the Author):

Overall, the authors have written an interesting paper that outlines the issue and provides a strong conceptual model that others can readily built upon. However, I think the manuscript can be significantly improved particularly the discussion section where authors discussed assembly processes (L 266 - 274). This section really lacks any discussion; it's a rehash of the results. Authors didn't provide any explanation for dominance of deterministic processes in communities with lower taxonomic diversity. Some previous studies have suggested that communities with smaller populations and/or lower diversity are more susceptible to drift (Vellend et al., 2014 and Evans et al. 2017). Please discuss your results in light of the reasons previous studies have suggested and then where this work fits into that.

Furthermore, I would urge the authors to consider making their analysis code public by including it in their supplemental. As it currently stands, it is unlikely that others could replicate their findings and build on their results without access to this code. I'm specifically mentioning of the β NTI analysis.

Specific comments

L28 Replace 'Nearest taxon' with 'Phylogenetic null modeling'

L404-405 In Picante you can only calculate β MNTD not β NTI. For calculating β NTI, you need to use a previously described custom scripts (Stegen, J. C., Lin, X. J., Konopka, A. E., & Fredrickson, J. K. (2012). Stochastic and deterministic assembly processes in subsurface microbial communities. *ISME Journal*, 6, 1653–1664). Please correct this.

Response to the reviewers' comments

Dear editor,

Thank you and these two reviewers for the valuable comments and suggestions regarding our manuscript (Manuscript ID: NCOMMS-18-26233B) entitled “**Diversity-triggered deterministic bacterial assembly constrains community functions**”. All the authors have discussed these comments and revised the manuscript accordingly. Point-by-point responses to the reviewers' comments are listed below. All changes made in the revised manuscript were highlighted with colored text.

Responses to reviewer #1:

Q1-1: No ecosystem function measures were made. Title, abstract and the manuscript text should refer to community function. Also recommend replacing microbiome with bacteria in the title.

Response: Thanks for the suggestion. The title has been revised as “**Diversity-triggered deterministic bacterial assembly constrains community functions**”. The “ecosystem function” in abstract and the manuscript context has been replaced by “community function”.

Q1-2: Line 29-31: These linkages are only correlative. Functional genes were not annotated to taxonomy, the authors should highlight that these are only correlations. Devoted is not the best term to use here.

Response: Thanks for the suggestion. We have highlighted the correlation. The term “Devoted” has been replaced. This sentence has been revised as “**However, in low-diversity communities, deterministic assembly processes were dominant associating with the reduction of specialized functions that are highly correlated to specific bacterial taxa.**”

Q1-3: Line 31-33: Finding a functional gene in the community does not indicate ecosystem consequences. A metagenome provides a snapshot of potential function of the community. Better use the term community function or such. Again, these are all correlative linkages so recommend using words like “probably” or “likely” so that readers are not misled.

Response: Thanks for the suggestion. The term “ecosystem function” has been replaced by “community function”. The recommended word “probably” was used in the revised sentence to highlight the correlative linkages. This sentence has been changed as “**Overall, we suggest that the**

low-diversity-induced deterministic community assembly processes probably constrain community functions, highlighting the potential roles of specialized functions in community assembly and in generating and sustaining the function of soil ecosystems.”

Q1-3: Line 147: carried out

Line 154, 156, 165, 167: Why ~ to show range?

Line 206: on dilution

Line 213: have suggested

Line 216: are usually

Line 228: occurs

Line 318: Taxonomic diversity?

Line 323: At what point?

Line 324: soil community potential functions were assessed in this study

There are still a lot of typos and grammatical errors, I've highlighted some here but I recommend careful review for language errors.

Response: Thanks for these corrections, which were all corrected. Moreover, the revised manuscript has been improved and grammatically checked by an English native speaker to improve the language.

Responses to reviewer #3:

Q3-1: Overall, the authors have written an interesting paper that outlines the issue and provides a strong conceptual model that others can readily built upon. However, I think the manuscript can be significantly improved particularly the discussion section where authors discussed assembly processes (L 266 - 274). This section really lacks any discussion; it's a rehash of the results. Authors didn't provide any explanation for dominance of deterministic processes in communities with lower taxonomic diversity. Some previous studies have suggested that communities with smaller populations and/or lower diversity are more susceptible to drift (Vellend et al., 2014 and Evans et al. 2017). Please discuss your results in light of the reasons previous studies have suggested and then where this work fits into that.

Response: Thank you for the positive comments of this manuscript. The discussion section on assembly processes has been improved. Related references have been cited and compared to

discuss our results. This section has been revised as “Our results showing the shift patterns of β NTI values through different pH and dilution levels clearly indicate that the community assembly tends to be dominated by deterministic processes when the taxonomic diversity level is low. This tendency was further confirmed by calculating the relationships between $|\beta$ NTI| values and bacterial Shannon diversity indices. Previous studies^{46,47} have suggested that the community is more susceptible to drift (stochastic process) or founder effects with lower biomass and smaller population. This is true at the period of low population or community size⁴⁸. However, in an established community with saturated population or community size, the local dominance of stochastic or deterministic processes can be strongly affected by dispersal, which depends on the local environment⁴⁹. Additionally, we found the deterministic processes were correlated with decreased functional diversity and relative abundances of specialized functions. Hence, we suggest that dispersal in low diversity communities may enhance environmental selection and contribute to deterministic (selection) processes. Therefore, there is a transition in bacterial community assembly from stochastic to deterministic processes with the decreasing of soil bacterial species richness and functional diversity. Moreover, according to the β NTI values, deterministic processes tend to be variable selection (β NTI > 2) at neutral pH and homogeneous selection (β NTI < -2) under acidic and alkaline conditions.”

Q3-2: Furthermore, I would urge the authors to consider making their analysis code public by including it in their supplemental. As it currently stands, it is unlikely that others could replicate their findings and build on their results without access to this code. I’m specifically mentioning of the β NTI analysis.

Response: Thanks for the suggestion. The R codes used for calculating β NTI metrics have been provided in the supplemental material.

Q3-3: L28 Replace ‘Nearest taxon’ with ‘Phylogenetic null modeling’

Response: Thanks for the suggestion. This sentence has been revised as “Phylogenetic null modeling analyses suggested that stochastic assembly processes were dominant in high-diversity communities.”

Q3-4: L404-405 In Picante you can only calculate β MNTD not β NTI. For calculating β NTI, you need to use a previously described custom scripts (Stegen, J. C., Lin, X. J., Konopka, A. E., &

Fredrickson, J. K. (2012). Stochastic and deterministic assembly processes in subsurface microbial communities. *ISME Journal*, 6, 1653–1664). Please correct this.

Response: Thanks for the comment. This section has been revised as “To infer the community assembly processes, we first calculated the mean nearest taxon distance metric using the picante R package ⁷⁰ and then implemented a previously developed null modeling approach to calculate the β -nearest taxon index (β NTI) ^{71,72}.” The R codes used for calculating β NTI have been provided in the supplemental material.